

# Evanescent acoustic-gravity modes in the isothermal atmosphere: systematization, applications to the earth's and solar atmospheres

Oleg K. Cheremnykh, Alla K. Fedorenko, Evgen I. Kryuchkov, Yuriy A. Selivanov

Space Research Institute NASU-SSAU, Kyiv, 03187, Ukraine

*Correspondence to:* Yuriy A. Selivanov (yuraslv@gmail.com)

**Abstract.** The objects of research in this work are evanescent wave modes in a gravitationally stratified atmosphere and their associated pseudo-modes. Whereas the former, according to the dispersion relation, rapidly decrease with distance from a certain surface, the latter, having the same dispersion law, differ from the first by the form of polarization and the character of their decreasing away from the surface. Within a linear hydrodynamic model, the propagation features of evanescent wave
modes in an isothermal atmosphere are studied. Research carried out for different assumptions about the properties of the medium. On this way, a new wave mode - anelastic evanescent wave mode - was discovered. Also, the possibility of the existence of a pseudo-mode related to it is indicated. The case of two isothermal media differing in temperature at the interface is studied in detail. It is shown that a non-divergent pseudo-mode with the dispersion of solar *f*-mode type can be realized on the interface for the specified horizontal scale. The newly discovered dispersion relation, at the interface of two
media, is satisfied by the wave mode, which has different types of amplitude versus height dependencies at different horizontal scales. The applicability of the obtained results to clarify the properties of *f*-mode observed on the Sun is analyzed.

Keywords: acoustic-gravity waves, evanescent wave modes, isothermal atmosphere, solar atmosphere, earth's atmosphere

**1 Introduction**

Acoustic - gravity waves (AGWs) in the Earth's atmosphere are studied theoretically and experimentally for more than 60 years. The linear theory of AGW (Hines, 1960; Yeh and Liu, 1974; Francis, 1975) admits the existence in the atmosphere of a continuous spectrum of freely propagating waves, consisting of acoustic and gravity regions, as well as of evanescent modes, which can only propagate horizontally.

The freely propagating AGWs effectively transfer the energy and momentum between various atmospheric layers and thus play an important role in the dynamics and energy balance of the atmosphere. These waves are generated by various sources (both natural and technogenic ones), which are accompanied by a significant energy output into the atmosphere. Further, when the AGWs propagate upward the energy conservation compensates for the decrease of the atmospheric density with the height by exponentially increasing amplitude. Therefore at a certain height the waves become nonlinear. Significant



progress in the development of the nonlinear theory of AGW was achieved by a number of authors, in particular, Belashov (1990), Nekrasov et al. (1995), Kaladze et al. (2008), Stenflo and Shukla (2009), Huang et al. (2014). Numerical modeling of the freely propagating AGWs in the realistic viscous and heat-conducting atmosphere is an important area of modern studies of these waves (i.e. Vadas, 2012; Cheremnykh et al., 2010).

Satellite observations of AGWs in the Earth's polar thermosphere indicate a prevailing presence of waves with oscillation periods concentrated around the Brunt–Väisälä period and of horizontal scale of about 500 - 700 km (Johnson et al., 1995; Innis and Conde, 2002; Fedorenko et al., 2015). Azimuths of the propagation of these AGW demonstrate the close connection with the directions of background winds in the thermosphere. Moreover, the amplitudes of the waves depend on the speed of headwind but do not depend on height (Fedorenko and Kryuchkov, 2013; Fedorenko et al., 2018).

These experimental results cannot be sufficiently explained by the theory of freely propagating AGWs. They may indicate waveguide or evanescent (along a horizontal surface) propagation of at least part of the observed waves.

As well as freely propagating AGWs, evanescent wave modes also play an important role in atmospheric dynamics of the Sun and planets. Evanescent waves propagate horizontally in an atmosphere, vertically stratified by gravity, subject to the presence of vertical gradients of parameters. The energy of these waves should decrease both up and down from the level

at which they are generated. Therefore, evanescent waves are most effectively generated in areas of presence of significant vertical gradients of temperature and density or strong local currents. For example, in the solar atmosphere suitable conditions for realization evanescent modes occur at the boundary between the chromosphere and corona (Jones, 1969). In the Earth's atmosphere, such waves can be efficiently generated at sharp vertical temperature gradients, for example, at the base of the thermosphere or at the heights of the tropo- and mesopause. Also, evanescent wave modes can emerge in the

presence of strong inhomogeneous winds, for example, in the region of the polar circulation of the thermosphere.

The study of evanescent waves traditionally gets less attention than the study of freely propagating AGWs. The most known of them are the horizontal Lamb wave and vertical oscillations with Brunt–Väisälä (BV) frequency (Beer, 1974; Waltercheid and Hecht, 2003). In hydrodynamics, physics of terrestrial and solar atmosphere, the surface gravity mode with dispersion $\omega^2 = k_x g$ is also well studied (Tolstoy, 1963; Jones, 1969). In particular, it was shown that it is the fundamental

mode ($f$ - mode) of oscillations in the solar atmosphere (Jones, 1969). Experimental $f$ - mode observations are used to study flows, refinement of the solar radius and other parameters of the Sun (Ghosh et al., 1995; Antia,1998). In the earth's atmosphere, evanescent waves are often observed at altitudes near the mesopause using ground-based instrumentation (Shimkhada et al., 2009).

In this paper, different types of evanescent acoustic-gravity modes characteristic of an isothermal atmosphere are

investigated using a set of linearized hydrodynamic equations. In particular, the possibility of the existence of a new type of evanescent acoustic-gravity modes with the dispersion $\omega^2 = k_x g(\gamma - 1)$ is proved. Also the possibility of realization the evanescent modes in the model of a thin temperature gap studied.





## 2 Evanescent modes in the isothermal atmosphere modes in the isothermal atmosphere

Consider an unbounded ideal isothermal atmosphere, stratified in a field of gravity. Linear perturbations in such a medium satisfy a set of four first order hydrodynamic equations (Hines, 1960). These equations are convenient to bring to a set of two second order equations for the perturbations of the horizontal $V_x$ and vertical $V_z$ particle velocities (Tolstoy, 1963):

$$\rho_0 \frac{\partial^2 V_x}{\partial t^2} = -\rho_0 g \frac{\partial V_z}{\partial x} + \frac{\partial}{\partial x}\left[\rho_0 c^2 \left(\frac{\partial V_x}{\partial x} + \frac{\partial V_z}{\partial z}\right)\right] , \tag{1}$$

$$\rho_0 \frac{\partial^2 V_z}{\partial t^2} = \rho_0 g \frac{\partial V_x}{\partial x} + \frac{\partial}{\partial z}\left[\rho_0 c^2 \left(\frac{\partial V_x}{\partial x} + \frac{\partial V_z}{\partial z}\right)\right] , \tag{2}$$

where $\rho_0$, $\gamma$, $g$ denote background atmosphere density, ratio of specific heats, acceleration of gravity, respectively; $c = \sqrt{\gamma g H}$ is the sound speed, $H = -\rho_0 / (d\rho_0/dz)$ is the density scale height.

Solutions to the system (1), (2) are usually searched for in the form:

$$V_x, V_z \sim \exp(az)\exp[i(\omega t - k_x x)] , \tag{3}$$

where $\omega$, $k_x$ are cyclic frequency and horizontal component of the wave vector, respectively; parameter $a$ sets the vertical scale of the change in the amplitude of velocities, $V_x$ and $V_z$, with the height, $z$. For brevity, we will refer to $a$ as the stratification of the corresponding mode.

The system (1), (2) allows, on the plot "frequency-wave number", for the existence of gravity and acoustic regions of

freely propagating waves, for which $a = \frac{1}{2H} \pm ik_z$ (Hines, 1960), where $k_z$ is the vertical component of the wave vector.

Also, from (1), (2) we get the solutions in the form of evanescent wave modes having real $a$ and propagating horizontally (Waltercheid and Hecht, 2003). Solutions in the form of evanescent modes are usually obtained by imposing additional conditions on the perturbation properties.

### 2.1 Non - divergent and and pseudo - non - divergent pseudo - non - divergent modes

Let us note the well known in hydrodynamics approximation of perturbations incompressibility (see, e.g., Ladikov-Roev et al., 2010), for which

$$div\vec{V} = \frac{\partial V_x}{\partial x} + \frac{\partial V_z}{\partial z} = 0 . \tag{4}$$

In frames of this approximation, we obtain the following equations from (1), (2):

$$\frac{\partial^2 V_x}{\partial t^2} = -g \frac{\partial V_z}{\partial x} , \tag{5}$$



$$\frac{\partial^2 V_z}{\partial t^2} = g\,\frac{\partial V_x}{\partial x} \quad. \tag{6}$$

After substituting (3) into equations (5), (6) we find:

$$-\omega^2 V_x = ik_x g V_z \quad,$$

$$-\omega^2 V_z = -ik_x g V_x \quad.$$

This yields a dispersion equation for incompressible wave modes in the form

$$\omega^2 = k_x g \quad. \tag{7}$$

Given the dispersion found, we obtain an expression for the polarization of the incompressible modes:

$$V_z = iV_x \quad. \tag{8}$$

Further, from the condition (4) and polarization (8) we get $a = k_x$. Insofar as $a$ is real value then non - divergent (ND)

wave mode has no periodic vertical solution and is horizontally propagating.

     Let us show that the dispersion relation (7) is also satisfied by another wave mode. After using this relation in (1), (2)

we get:

$$V_x\big(\gamma H k_x - 1\big) - iV_z\big(1 - \gamma a H\big) = 0 \quad, \tag{9}$$

$$iV_x\big(1 - \gamma H a - \gamma\big) - V_z\left(1 + \frac{\gamma H a^2}{k_x} - \frac{\gamma a}{k_x}\right) = 0 \quad. \tag{10}$$

From the system (9), (10) follows:

$$a^2 - \frac{a}{H} + \frac{k_x}{H}\big(1 - k_x H\big) = 0 \quad,$$

which implies that there are two solutions to this equation:

$$a_{1,2} = \frac{1}{2}\left[\frac{1}{H} \pm \left(\frac{1}{H} - 2k_x\right)\right] = \begin{cases} \dfrac{1}{H} - k_x \\ k_x \end{cases} \quad. \tag{11}$$

     The lower solution in (11) corresponds to the non-divergent (ND) wave mode, and the upper one we call pseudo-non-

divergent mode (NDp). The expression for polarization NDp is obtained from (9) and has the form:

$$V_x\left(\frac{1}{\gamma H} - k_x\right) = -i\left(k_x - \frac{\gamma - 1}{\gamma H}\right)V_z \quad.$$

Also for this mode holds the equation



$$div\vec{V} = \frac{V_z}{H}\frac{(1-2k_xH)}{(1-\gamma k_xH)} \ ,$$

which shows that for NDp mode $div\vec{V} = 0$ only when $k_x = 1/2H$ .

## 2.2 Anelastic and pseudo-anelastic modes

Let us show that equations (1), (2) indicate that another wave mode, not previously studied, may exist. To do this, we

introduce, according to Bannon (1996), the anelastic linear perturbations, which satisfy the condition

$$div(\rho_0\vec{V}) = 0 \ . \tag{12}$$

In the isothermal atmosphere with barometric density distribution we have

$$\frac{\partial\rho_0}{\partial z} = -\frac{\rho_0}{H} ,$$

therefore, for such *anelastic* perturbations, the following equation holds:

$$div\vec{V} = \frac{V_z}{H} \ . \tag{13}$$

Substituting (13) into equations (1), (2) we get:

$$\frac{\partial^2 V_x}{\partial t^2} = g(\gamma-1)\frac{\partial V_z}{\partial x} \ ,$$

$$\frac{\partial^2 V_z}{\partial t^2} = -g(\gamma-1)\frac{\partial V_x}{\partial x} \ .$$

Thus, given (3), should:

$$\omega^2 V_x = ik_x g(\gamma-1)V_z \ , \tag{14}$$

$$\omega^2 V_z = -ik_x(\gamma-1)gV_x \ . \tag{15}$$

Then the dispersion equation for *anelastic* (AE) *modes* takes the form:

$$\omega^2 = k_x g(\gamma-1). \tag{16}$$

With the resulting dispersion, polarization follows from equations (14), (15):

$$V_x = iV_z \ . \tag{17}$$





Further, taking into account (13), we obtain $a = \dfrac{1}{H} - k_x$. Consequently, the AE mode also does not have a solution periodic

vertically and can only propagate horizontally.

After substituting the dispersion (16) into equations (1), (2) we get:

$$V_x\left(1 - \gamma + \gamma H k_x\right) - iV_z\left(1 - \gamma a H\right) = 0 \ , \tag{18}$$

$$5 \quad iV_x\left(1 - \gamma + \gamma H a\right) + V_z\left(1 - \gamma - \frac{\gamma H a^2}{k_x} + \frac{\gamma a}{k_x}\right) = 0 \ , \tag{19}$$

whence we get a pair of values $a$, identical to (11). Consequently, there is another wave solution that satisfies the equation
(16), we call it *pseudo-anelastic* (AEp) *mode*. The upper value in (11) corresponds to the AE wave mode, and the lower –
AEp. Polarization of the AEp mode has the form:

$$V_x\left(k_x - \frac{\gamma - 1}{\gamma H}\right) = i\left(\frac{1}{\gamma H} - k_x\right)V_z$$

10 that follows from (18) or (19).

## 3 General properties of evanescent modes

Let us prove that the different types of evanescent modes characteristic of an isothermal atmosphere are related. We
substitute (3) into system (1), (2) without additional conditions that were imposed in Section 2 when deriving ND and AE
mode. As a result, we get:

$$15 \quad V_z\left(a - \frac{g}{c^2}\right) - ik_x V_x\left(1 - \frac{\omega^2}{k_x^2 c^2}\right) = 0 \ , \tag{20}$$

$$V_x\left(a - \frac{N^2}{g}\right) - ik_x V_z\left(\frac{N^2}{\omega^2} - 1\right) = 0 \ , \tag{21}$$

where $N^2 = \dfrac{g}{H}\dfrac{(\gamma - 1)}{\gamma}$ is the square of the Brunt-Väisälä frequency.

From system (20), (21) we obtain the dispersion equation:

$$\left(\frac{1}{\gamma H} - a\right)\left(a - \frac{\gamma - 1}{\gamma H}\right) = k_x^2\left(\frac{N^2}{\omega^2} - 1\right)\left(1 - \frac{\omega^2}{k_x^2 c^2}\right) \ . \tag{22}$$





Expressions $\omega^2 = N^2$ and $\omega^2 = k_x^2 c^2$ are well-known dispersions of Brunt-Väisälä oscillations with $a = 1/\gamma H$ and Lamb waves (L) with $a = (\gamma - 1)/\gamma H$. In addition to these known modes, dispersion (22) also admits the existence of additional solutions in the form of BV pseudo-modes (BVp) with $\omega^2 = N^2$, $a = (\gamma - 1)/\gamma H$ and Lamb pseudo-modes (Lp) with $\omega^2 = k_x^2 c^2$, $a = 1/\gamma H$ (Beer, 1974; Waltercheid and Hecht, 2003).

Then represent (22) in the form of a quadratic equation with respect to $a$:

$$a^2 - \frac{a}{H} + \frac{\omega^2}{c^2} - k_x^2 + \frac{k_x^2 g^2}{\omega^2 c^2}(\gamma - 1) = 0 \ .$$

The solution to this equation is:

$$a = \frac{1}{2H} \pm \sqrt{\frac{(k_x g - \omega^2)(\omega^2 - k_x g(\gamma - 1))}{\omega^2 c^2} + \left(k_x - \frac{1}{2H}\right)^2} \ . \tag{23}$$

From which it follows that for modes with dispersions $\omega^2 = k_x g(\gamma - 1)$ and $\omega^2 = k_x g$ there are two possible values: $a = k_x$

and $a = \frac{1}{H} - k_x$. The first value corresponds to modes ND and AEp, and the second – to NDp and AE.

Thus, each evanescent mode can be associated with a pseudo-mode, which satisfies the same dispersion relation, but differs in polarization and dependence of the amplitude from the height, i.e., in its stratification. Table 1 presents the properties of different evanescent modes, characteristic of the isothermal atmosphere: BV oscillations, Lamb waves, non-divergent and anelastic modes, along with associated pseudo-modes: BVp, Lp, NDp, AEp. Table 1 shows that for all pseudo-

modes, the polarization changes depending on the value of $k_x$. Wave modes AE and ND at $k_x = 1/2H$ completely coincide with AEp and NDp, respectively.

The location of the dispersion curves for anelastic and non-divergent modes relative to gravity and acoustic regions in the ($\omega$, $k_x$) plane is shown in Fig. 1. The $\omega^2 = k_x g(\gamma - 1)$ mode touches the gravity region of freely propagating AGWs at the same value $k_x = 1/2H$ at which the $\omega^2 = k_x g$ curve touches the acoustic region (see Figure 1). In this case, the dispersion

curves of AE and ND modes are symmetric relative to the "characteristic" curve (see Beer, 1974), which separates the AGW acoustic region from the AGW gravity region. In fact, the characteristic curve is the geometric mean of the dispersion curves of AE and ND modes with $\omega^2 = \sqrt{k_x^2 g^2 (\gamma - 1)} = N k_x c$ .

From Figure 1 we see that the dispersion curves of different evanescent modes have intersection at separate points. Lamb dispersion curve with $\omega^2 = k_x^2 c^2$ intersects with BV curve with $\omega^2 = N^2$ at the point $k_x = N/c$. However, these

modes cannot interact with each other by reason of different polarizations and values of $a$. At the same time, the pairs Lp - BV and L - BVp completely coincide in these properties and are indistinguishable at the intersection points.



Dispersion curves $\omega^2 = k_x g$ and $\omega^2 = k_x g(\gamma - 1)$ intersect with the Lamb curve and the BV curve at points $k_x = 1/\gamma H$ , $k_x = (\gamma - 1)/\gamma H$ . In addition, the ND mode curve intersects with the Lamb curve at the same value $k_x$ , at which the AE mode curve intersects with the BV curve (see Fig. 1). ND and AE modes cannot interact with the Lamb mode and BV oscillations due to different polarizations (Table 1). Pseudo-modes NDp and AEp, at the points of intersection with the Lamb wave and the BV oscillations, have the same polarization and values of $a$ . Similarly, ND and AE are indistinguishable at the points of intersection with Lp and BVp. Table 2 shows all evanescent modes that coincide with each other at the points of intersection of the dispersion curves, and between which interaction is possible. The cases of ND and AE mode curves intersection with curves ($a = 1/2H$), which separate the area of freely propagating AGWs from the evanescent area, are not presented in Table 2.

## 4 The energy of evanescent modes in an isothermal atmosphere

In Sections 2 and 3, we considered a model of an unlimited isothermal stratified atmosphere to determine which types of evanescent modes can satisfy the initial system of equations (1), (2). However, in an infinitely extended medium, the necessary condition for the existence of evanescent modes is the absence of unlimited growth of oscillation energy above and below the height level at which they are generated. It is easy to verify that in an isothermal infinite atmosphere, none of the modes listed in Table 1 satisfy this condition.

Suppose further that an evanescent wave is generated at a certain altitude level $z = 0$ . The kinetic energy density $E \sim \rho(z)\left(V_x^2 + V_z^2\right)$ of waves should decrease both up and down from the level $z = 0$ . When $z \to +\infty$ the energy density $E \sim \exp\left(2a - \dfrac{1}{H}\right)z \to 0$ , if $a < 1/2H$ , and $E \to \infty$ , if a $a > 1/2H$ . When $z \to -\infty$ the energy density $E \to 0$ , if a $a > 1/2H$ and $E \to \infty$ , if a $a < 1/2H$ . Based on these considerations, it is not difficult to understand how the energy density varies with height for different types of evanescent modes in an infinite isothermal atmosphere (see Table 3). Therefore, for the realization of such modes, it is necessary to have boundaries in the medium at which the condition for reducing energy in both directions from this boundary can be satisfied.

The presence of boundaries is not the only condition that can limit the energy of the evanescent mode. If the equality $a = 1/2H$ holds for these modes, then their energy is not varies with height in an isothermal atmosphere. For an infinite atmosphere, this solution does not seem to be physical, but it can make sense for a real atmosphere of finite height. As follows from (11), for the ND and AE modes, as well as their pseudo-modes, the condition $a = 1/2H$ performed at the point $k_x = 1/2H$ . Also, at this point, the ND mode is identical to the NDp mode, and the AE mode completely coincides with AEp. In addition, when $k_x = 1/2H$ these evanescent modes adjoin the border of regions of freely propagating AGW (see Fig. 1).



Consider some features of the energy balance for the evanescent modes. It follows from equation (20) that:

$$|V_z|^2\left(a-\frac{g}{c^2}\right)^2 = k_x^2|V_x|^2\left(1-\frac{\omega^2}{k_x^2 c^2}\right)^2 . \tag{24}$$

Combining equations (22) and (24) gives the relation:

$$\rho_0|V_x|^2\left(1-\frac{\omega^2}{k_x^2 c^2}\right)\left(a-\frac{N^2}{g}\right) = \rho_0|V_z|^2\left(\frac{N^2}{\omega^2}-1\right)\left(\frac{g}{c^2}-a\right) . \tag{25}$$

5    The average density of the kinetic energy of the perturbations is $E_k = \frac{1}{4}\rho_0\left(V_x^2 + V_z^2\right)$ and of the potential energy is

$$E_p = \frac{1}{4}\rho_0\left(V_x^2\frac{\omega^2}{k_x^2 c^2} + V_z^2\frac{N^2}{\omega^2}\right)$$ (Yeh, Liu, 1974; Fedorenko, 2010). Therefore, from equation (25) it follows that for the

evanescent modes $E_k \neq E_p$. At the same time, for freely propagating AGWs is always fulfilled the equality $E_k = E_p$

(Yeh, Liu, 1974). At the point $a = 1/2H$ where evanescent modes on the plane ($\omega$, $k_x$) in Fig. 1 are adjacent to areas of

freely propagating AGWs, the equality $a - \frac{N^2}{g} = \frac{g}{c^2} - a$ holds. Taking this circumstance into account, from (25) we obtain:

10    $$\frac{\rho_0}{4}\left(|V_x|^2 + |V_z|^2\right) = \frac{\rho_0}{4}\left(|V_x|^2\frac{\omega^2}{k_x^2 c^2} + |V_z|^2\frac{N^2}{\omega^2}\right) , \tag{26}$$

that is, at this point $E_k = E_p$.

## 5 Evanescent modes at the interface of isothermal media

Let us consider the possibility of realization of evanescent modes in the atmosphere at a thin interface between two

isothermal half-spaces of infinite extent, which differ in temperature $T$. Let the boundary be localized at some altitude level

15    $z = 0$. In the lower half-space ($z < 0$) we have $T = T_1$, while in the upper half-space ($z > 0$) we have $T = T_2$ and it is

assumed that $T_2 > T_1$. Note that a similar model was considered by Rosental and Gough (1994). We will search for solutions

to the system (1), (2) in the form of $V_x, V_z \sim \exp(a_1 z)\exp[i(\omega t - k_x x)]$ for the lower half-plane and in the form

$V_x, V_z \sim \exp(a_2 z)\exp[i(\omega t - k_x x)]$ for the upper half-plane. Substituting these dependencies into (1), (2) yields:

$$a_1 = \frac{1}{2H_1} \pm \left(\frac{1}{4H_1^2} - \frac{\omega^2}{c_1^2} + k_x^2 - k_x^2\frac{N_1^2}{\omega^2}\right)^{1/2} , \tag{27}$$




$$a_2 = \frac{1}{2H_2} \pm \left( \frac{1}{4H_2^2} - \frac{\omega^2}{c_2^2} + k_x^2 - k_x^2 \frac{N_2^2}{\omega^2} \right)^{1/2} .$$ (28)

Here indices 1 and 2 denote the values in the lower and upper half-spaces, respectively.

The density of the kinetic energy of evanescent waves should decrease from the level $z = 0$ both up and down. This condition limits the possible values of $a_1$ and $a_2$. In the upper half-space ( $z > 0$ ), when $z \to +\infty$ the energy density

$E_2 \sim \exp\left( 2a_2 - \frac{1}{H_2} \right) z \to 0$ , if $a_2 < 1/2H_2$ . In the lower half-space ( $z < 0$ ), when $z \to -\infty$ the energy density

$E_1 \sim \exp\left( 2a_1 - \frac{1}{H_1} \right) z \to 0$, if $a_1 > 1/2H_1$. Therefore, it is necessary to take in the expression (27) for $a_1$ the solution with

a "+" sign  and in expression (28) for $a_2$, with a "-" sign, so that the energy decreases on both sides of the interface.

It is also necessary to consider that the possible values of $a_1$ and $a_2$ must satisfy the boundary condition (Tolstoy,1963; Rosental and Gough, 1994),  arising from (1), (2):

$\rho_1 c_1^2 \left. \frac{g k_x^2 - \omega^2 a_1}{\omega^2 - c_1^2 k_x^2} \right|_{z=-0} = \rho_2 c_2^2 \left. \frac{g k_x^2 - \omega^2 a_2}{\omega^2 - c_2^2 k_x^2} \right|_{z=+0} ,$ (29)

where $\rho_1$ and $\rho_2$ are the densities on both sides of the boundary. The procedure for deriving equality (29) is exactly the same as in the papers by Cheremnykh et al. (2018a) and Cheremnykh et al. (2018b).  In the barometric atmosphere we have $\rho c^2 = \gamma p_0$, where $p_0$ is the equilibrium pressure, which must be continuous in the transition through the boundary. Therefore, when $\gamma_1 = \gamma_2$ equation (29) can be written as:

$\frac{g k_x^2 - \omega^2 a_1}{\omega^2 - c_1^2 k_x^2} = \frac{g k_x^2 - \omega^2 a_2}{\omega^2 - c_2^2 k_x^2}$ . (30)

Dispersion dependencies of $\omega = f(k_x)$ calculated numerically by means of the expression (30) are shown in Fig. 2a for different values of the parameter $d = H_2 / H_1$ . On each of these curves, the condition for decreasing energy up and down from the interface is satisfied.  The long-wavelength part of the spectrum, where the most interesting features appear, is shown in more detail in Fig. 2b. Also shown in these figures are the dispersion curves $\omega = \sqrt{k_x g}$ and $\omega = \sqrt{k_x g(\gamma - 1)}$ for

the ND and AE wave modes. The discontinuities of the $\omega = f(k_x)$ curves, as well as their cut-off for smaller $k_x$ values, are due to requirements $a_1 > 1/2H_1$ and $a_2 < 1/2H_2$ . Some features of the behaviour of $\omega = f(k_x)$ will be discussed below.





As shown by Miles and Roberts (1992), the dispersion equation (30) can be rewritten to a polynomial form suitable for analysis:

$$\omega^8 - 2c_1^2(d+1)k_x^2\omega^6 + \left[c_1^4(d+1)^2k_x^4 + (2\gamma-1)k_x^2g^2\right]\omega^4 - 2(\gamma-1)c_1^2(d+1)k_x^4g^2\omega^2 - c_1^4(d-1)^2k_x^6g^2 = 0 \,. \tag{31}$$

Non-physical solutions (Miles and Roberts, 1992) arising from quadratic expressions under the radicals were excluded from consideration while obtaining equation (31) (see (27), (28)). Expressions (31) can be analyzed by examining their asymptotic behavior.

If $k_x^2c_1^2 \gg \omega^2$, then from (31) we get:

$$\omega^4 - \frac{2N_1^2}{d+1}\omega^2 - \frac{(d-1)^2}{(d+1)^2}k_x^2g^2 \approx 0 \,.$$

It follows from this expression:

$$\omega^2 = \frac{1}{d+1}\left[N_1^2 + \sqrt{N_1^4 + (d-1)^2k_x^2g^2}\right] \,. \tag{32}$$

The expression (32) contains an interesting dependence of the frequency on the parameter $d$. In the limit $d \to \infty$, the dispersion $\omega^2 \approx k_x g$ of the ND (NDp) mode, independent of the properties of both environments, follows from (32). With $d \to 1$ and using (32), we obtain the dispersion of the BV (BVp) mode with the parameters of the lower medium, that is, $\omega^2 \approx N_1^2$. The indicated asymptotic features are visible on the curves shown in Fig. 2 below.

In the long-wave limit, i.e., at $k_x \to 0$, from (31) it follows:

$$(2\gamma-1)\omega^4 - 2(\gamma-1)c_1^2(d+1)k_x^2\omega^2 - c_1^4(d-1)^2k_x^4 \approx 0 \,.$$

Hence we find:

$$\omega^2 = \frac{c_1^2k_x^2}{2\gamma-1}\left[(\gamma-1)(d+1) + \sqrt{\gamma^2(d+1)^2 - 4d(2\gamma-1)}\right] \,. \tag{33}$$

For the considered small $k_x$, for different values of d, from (33) we obtain the family of Lamb-type acoustic modes (see Fig. 2b). For large values of $d$, using (33), we obtain the expression $\omega^2 \approx c_1^2k_x^2d = c_2^2k_x^2$, i.e. the oscillation frequency is determined by the characteristics of the medium in the upper half-space.

The evanescent modes frequencies lie on the $(\omega, k_x)$ plane between the acoustic and gravity regions of freely propagating AGW determined for upper and lower media separately (see Fig. 1). It is necessary to take into account when considering evanescent modes at the boundary of two isothermal media with different temperatures, that the evanescent



regions are different in the upper and lower half-planes. On the $(\omega, k_x)$ plane, these regions are shifted relative to each other the more, the more is the value of $d$. At the same time, the wave modes at the interface of the media should remain evanescent in both media, and their dispersions should be enclosed within the overlap region of two evanescent regions. The cut-off curves for evanescent regions in the media under consideration are obtained in case of the null expressions under the

radicals in (27) and (28). Gaps on the $\omega = f(k_x)$ dispersion curves are due to the evanescent areas of the two media do not match (see. Fig. 3).

Note that the dispersion curves $\omega = f(k_x)$ for values $d \leq 4$ are mostly inside both evanescent regions (see Fig. 3 a, 3 b), except for the longest waves. When $d \geq 4$, the dispersion curve $\omega = f(k_x)$ breaks into two separate branches (see Fig. 3 c, 3 d). The long-wave branch is acoustic, and another branch with $k_x \geq 0.4 H_1$ is surface gravity by its physical nature.

## 6  Characteristic scales of ND and AE evanescent modes on the discontinuity

In an unlimited isothermal medium, evanescent modes are separate "pure" solutions of hydrodynamic equations. At the interface between two isothermal media with different temperatures, dispersion of the evanescent modes have a combined character, composing different types of "pure" modes, depending on the value of the parameter $d$ and spectral properties $\omega(k_x)$.

For some values of $d$, the curves of the dispersion equation (30) approach fairly close to the curves $\omega^2 = k_x g$ and $\omega^2 = k_x g(\gamma - 1)$, and also intersect them at different points. These intersection points correspond to the specific value of $k_x$, at which the dispersions of the ND and AE modes are realized, in the model under consideration, in a "pure" form. Let us now examine these cases in more detail. For this purpose, we substitute the dispersion relations $\omega^2 = k_x g$ and $\omega^2 = k_x g(\gamma - 1)$ directly into (27), (28), and then into the boundary condition (30).

As was shown in Section 2, for dispersion relations $\omega^2 = k_x g$ and $\omega^2 = k_x g(\gamma - 1)$, values $a_1$ and $a_2$ coincide and are determined by expressions (11). Consider the valid values of $a_1$ and $a_2$ for these dispersions with regard to the requirement of energy decay in both directions from the interface $a_1 > 1/2H_1$ and $a_2 < 1/2H_2$.

### 6.1 Dispersion of the form $\omega^2 = k_x g$

For a dispersion of the form $\omega^2 = k_x g$, we first analyze the stratification of the ND mode with $a_1 = k_x$, $a_2 = k_x$. In order

for the energy of this mode to decay in both directions from the discontinuity, the following inequalities $1/2H_1 < k_x < 1/2H_2$ must be satisfied, i.e., $H_1 > H_2$. Therefore, ND mode can be realized at the discontinuity, if the





ambient temperature in the upper region is less and the density is greater than they are in the lower region. This situation corresponds to the unstable state of the atmosphere (see Roberts, B., 1991).

Take the stratification of the NDp modes in the form of $a_1 = \dfrac{1}{H_1} - k_x$, $a_2 = \dfrac{1}{H_2} - k_x$. The energy in this case decreases both ways from the discontinuity, if $1/2H_2 < k_x < 1/2H_1$, i.e. when $H_2 > H_1$. This condition corresponds to the stable

state and the case under consideration. For the NDp mode from the dispersion equation (30) we get:

$$H_2\left(\frac{1}{\gamma H_2} - k_x\right)\left(2k_x - \frac{1}{H_1}\right) = H_1\left(\frac{1}{\gamma H_1} - k_x\right)\left(2k_x - \frac{1}{H_2}\right) \ , \ k_x \neq 1/\gamma H_1 \ , \ k_x \neq 1/\gamma H_2 \ . \tag{34}$$

From (34) it follows:

$$k_x = \frac{d+1}{4dH_1}\left(1 \pm \sqrt{1 - \frac{8d}{\gamma(d^2 - 1)}}\right) \ . \tag{35}$$

Figure 4a shows values of $k_x$ for which the dispersion curve $\omega^2 = k_x g$ intersects with the calculated dispersion curve

$\omega = f(k_x)$ depending on the parameter $d$. The upper solid curve in this figure corresponds to the solution (35) with the sign "+" before the radical and shows the points of intersection with the shorter wavelength branch. The lower dashed curve corresponds to the solution with a sign "- " and represents the points of intersection with the long-wavelength branch. For the upper curve $k_x \to 1/2H_1$ when $d \to \infty$. For $d < 2.5$, there are no intersections of the curve $\omega = f(k_x)$ calculated numerically from (30) with the curve for the dispersion $\omega^2 = k_x g$.

When combining the stratifications for ND modes as $a_1 = k_x$ and for NDp modes as $a_2 = \dfrac{1}{H_2} - k_x$, equation (30) yields the only possible value of $k_x = 1/2H_2$. For a combination of stratifications $a_1 = \dfrac{1}{H_1} - k_x$ (NDp) , $a_2 = k_x$ (ND) we get $k_x = 1/2H_1$. Both of these cases do not satisfy the condition of energy decrease with height.

Thus, consideration of the possible values of $a_1$ and $a_2$ leads to the conclusion that on the interface of two isothermal media with $H_2 > H_1$ can only be implemented NDp mode with a dispersion $\omega^2 = k_x g$ and a specific scale $k_x \sim 1/2H_1$.

## 6.2   Dispersion of the form $\omega^2 = k_x g(\gamma - 1)$

For the AE stratification of the form $a_1 = \dfrac{1}{H_1} - k_x$, $a_2 = \dfrac{1}{H_2} - k_x$ and for the AEp  stratification of the form $a_1 = k_x$, $a_2 = k_x$, from the dispersion equation (30) follows the identity $H_1 = H_2$. Therefore, such modes do not realize at a





temperature discontinuity. Apparently, to study the conditions of realization of AE and AEp modes, it is necessary to consider atmospheric models in which height profile $H(z)$ is continuous.

It should be noted that for the dispersion of the form $\omega^2 = k_x g (\gamma - 1)$, cases of combined modes stratifications are possible, satisfying the condition of decreasing energy on both sides of the boundary. So, for the combination of

stratifications $a_1 = k_x$ (AEp), $a_2 = \dfrac{1}{H_2} - k_x$ (AE) from (30) we obtain the relation:

$$ H_2 k_x (2 - \gamma) = H_1 \gamma \left( k_x - \frac{\gamma - 1}{\gamma H_1} \right) \ . $$

Whence $k_x = \dfrac{\gamma - 1}{H_1 [\gamma - (2 - \gamma) d]}$ . In this case, the inequality $d < \gamma / (2 - \gamma)$ must be satisfied. When $\gamma = 5/3$ we get the

following restriction: $d < 5$. Given this limitation and condition $k_x > 1/2 d H_1$, we obtain that a mode with a dispersion of

$\omega^2 = k_x g (\gamma - 1)$ and stratification of AE type for the upper half-space and of AEp type for the lower half-space can

propagate at the boundary in the range $1 < d < 5$ and for $k_x > 1/2 H_1$. For the stratifications $a_1 = \dfrac{1}{H_1} - k_x$ (AE), $a_2 = k_x$

(AEp) from equation (30) we obtain the relation:

$$ H_2 \gamma \left( k_x - \frac{\gamma - 1}{\gamma H_2} \right) = H_1 k_x (2 - \gamma) \ . $$

It implies the ratio $k_x = \dfrac{\gamma - 1}{H_1 [\gamma d - (2 - \gamma)]}$, in which the parameter $d$ can take any values with $d > 1$, and the horizontal wave

number is limited by the inequality $k_x < 1/2 H_1$. Features of the behavior of the $\omega^2 = k_x g (\gamma - 1)$ mode at the discontinuity,

depending on the scale $k_x$ are shown in Fig. 4b.

## 7   Discussion

Let us dwell on some of the results in terms of their use for the analysis of experimental data.

With the $f$-mode observed on the Sun, one should identify the mode that we classify as ND mode, for which $\omega^2 = k_x g$,

$V_z \sim \exp(k_x z)$ and $div \vec{V} = 0$ (Roberts, B., 1991). In the framework of the considered temperature discontinuity model, it

was shown that with $T_1 < T_2$ (corresponds to the chromosphere-corona interface) the condition for decreasing amplitude with

height to both sides of the interface is satisfied only by the NDp mode with $\omega^2 = k_x g$, $V_z \sim \exp\left( \dfrac{1}{H} - k_x \right) z$ and $div \vec{V} \neq 0$.

When the ratio $d \to \infty$ (i.e., $H_2 / H_1 \to \infty$), the NDp mode with $k_x \to 1/2 H_1$ asymptotically approaches ND mode. On the



interface between the chromosphere and the solar corona $d$ is large, but of finite magnitude: $d \sim 50$ (Athay, 1976; Jones, 1969). Therefore, the condition of the presence of a free surface, which is required for the realization of the ND mode, is fulfilled only approximately. Therefore, in the framework of the temperature discontinuity model, the $f$ - mode observed on the Sun should not be compared with the non-divergent ND mode, but with non-divergent *pseudo* – mode NDp.

For the Earth's atmosphere, the maximum possible value of $d$ is observed at the interface between the thermosphere with $T_2 \sim 800 - 1500K$ (depending on solar activity) and the underlying atmosphere with $T_1 \sim 300K$. When $d = 5$, the dispersion (30) asymptotically tends to $\omega^2 = k_x g (\gamma - 1)$ with $k_x \to \infty$. Therefore, it can be expected that evanescent modes in this case will be close to $\omega^2 = k_x g (\gamma - 1)$.

In other layers of the earth's atmosphere we have $d \le 1.3$ (Jursa, 1985). As follows from (33), for small values of
$d \le 1.3$ and for the wavelengths in the interval $k_x \sim (0.5 - 1.5) H_1$, the equality $\omega^2 \to N^2$ is satisfied (see Fig. 2). Therefore, it can be expected that at small positive temperature gradients in the atmosphere, waves with a frequency close to the frequency of Brent-Väisälä should prevail. These conclusions experimentally confirm (Shimkhada et al., 2009) the results of observations of short-period evanescent waves with small wavelengths at altitudes near the mesopause.

## 8  Main results

In the paper, different types of evanescent acoustic-gravity modes characteristic of an isothermal atmosphere are investigated. A new mode was derived in the form of anelastic acoustic-gravity wave mode with the dispersion equation $\omega^2 = k_x g (\gamma - 1)$. The main properties of the AE mode are presented in Table 1 in comparison with other known evanescent modes. It is shown that for both anelastic and non-divergent modes there are pseudo-modes that satisfy the same dispersions, but having different polarization and the dependence of the amplitude of the disturbances on the height.

For AE and ND evanescent modes, the value of $k_x \to 1/2H$ sets a special scale (wavelength) at which these modes are identical to their pseudo-modes AEp and NDp. In addition, at the same point they are adjacent to the boundaries of the continuous spectrum (AE mode to the gravity region, and ND mode to the acoustic region, respectively).

The features of the evanescent modes realization at the interface of two isothermal media are considered. It is shown that in this case, dispersions of evanescent modes are combined, merging the features of different types of modes
characteristic of an unlimited isothermal atmosphere. This effect is most pronounced in the following asymptotic cases: 1) when $d \to \infty$, we obtain the dispersion for the ND (NDp) mode in the form $\omega^2 \approx k_x g$; 2) when $d \to 1$, for scales $k_x \sim H_1$, a mode with $\omega^2 \approx N_1^2$ is realized; 3) for $k_x \to 0$, a Lamb wave with a dispersion relation of the form $\omega^2 \approx c_2^2 k_x^2$ is obtained, which depends only on the parameters of the medium in the upper half-space.


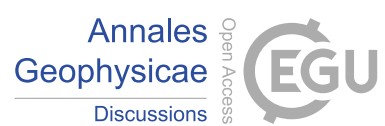

It was demonstrated that on the interface of two isothermal media with $T_2 > T_1$, the NDp mode with the dispersion $\omega^2 = k_x g$ and the selected scale $k_x \sim 1/2H_1$ is realized. At the same time, the ND mode does not satisfy the condition of decreasing energy on each side of the interface. Dispersion $\omega^2 = k_x g (\gamma - 1)$ on the interface of two media is satisfied by the wave mode, which has different types of amplitude versus height dependencies at different horizontal scales $k_x$. When $k_x > 1/2H_1$, the height dependence of AE amplitude for $z > 0$ and AEp amplitude for $z < 0$ satisfy the condition of decreasing energy from the interface. On the contrary, when $k_x < 1/2H_1$, this condition is satisfied by AEp amplitude for $z > 0$ and AE amplitude for $z < 0$.

It is important to note that according to our analysis in the framework of the temperature discontinuity model: (1) the $f$-mode observed on the Sun should not be compared with the non-divergent ( $\omega^2 = k_x g$, $divV = 0$ ) mode, but with its non-divergent *pseudo*–mode ( $\omega^2 = k_x g$, $divV \neq 0$ ). (2) At the interface between the earth's thermosphere and the underlying atmosphere it can be expected that evanescent modes with short wavelengths will be close to the new mode ( $\omega^2 = k_x g (\gamma - 1)$ ). (3) Oscillations with a frequency close to the frequency of Brent-Väisälä should prevail at altitudes near the earth's mesopause.

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



**Table 1: Properties of different evanescent acoustic-gravity modes**

| Mode type | Dispersion | $a$ | Polarization |
|---|---|---|---|
| Lamb Wave (L) | $\omega^2 = k_x^2 c^2$ | $\dfrac{\gamma - 1}{\gamma H}$ | $V_z = 0 \,;\, V_x \neq 0$ |
| Lamb's Pseudo-mode (Lp) | | $\dfrac{1}{\gamma H}$ | $V_x(2-\gamma)k_x g = i\left(N^2 - k_x^2 c^2\right)V_z$ |
| BV Oscillations (BV) | $\omega^2 = N^2$ | $\dfrac{1}{\gamma H}$ | $V_x = 0 \,;\, V_z \neq 0$ |
| BV Pseudo-mode (BVp) | | $\dfrac{\gamma - 1}{\gamma H}$ | $V_x\left(k_x^2 c^2 - N^2\right) = i(2-\gamma)k_x g V_z$ |
| Non-divergent (ND) mode, $div\vec{V} = 0$ | $\omega^2 = k_x g$ | $k_x$ | $V_x = -iV_z$ |
| Pseudo-non-divergent mode (NDp), $div\vec{V} \neq 0$ | | $\dfrac{1}{H} - k_x$ | $V_x\left(\dfrac{1}{\gamma H} - k_x\right) = -i\left(k_x - \dfrac{\gamma - 1}{\gamma H}\right)V_z$ |
| Anelastic mode (AE), $div\left(\rho_0\vec{V}\right) = 0$ | $\omega^2 = k_x g(\gamma - 1)$ | $\dfrac{1}{H} - k_x$ | $V_x = iV_z$ |
| Pseudo-anelastic mode (AEp), $div\left(\rho_0\vec{V}\right) \neq 0$ | | $k_x$ | $V_x\left(k_x - \dfrac{\gamma - 1}{\gamma H}\right) = i\left(\dfrac{1}{\gamma H} - k_x\right)V_z$ |





**Table 2: The coincidence of the evanescent mode properties at the intersection points of the dispersion curves \***

| L | Lp | BV | BVp | ND | NDp | AE | AEp |
|-----|-----|-----|-----|-----|-----|-----|-----|
| BVp | BV | Lp | L | Lp | L | Lp | L |
| NDp | ND | NDp | ND | BVp | BV | BVp | BV |
| AMp | AM | AMp | AM | | | | |

*Note.* The bottom rows show the modes that are indistinguishable from the corresponding mode of the top row at the point of intersection of the dispersion curves.



**Table 3: The change in energy density of evanescent modes with height in an infinite isothermal atmosphere**

|  | L | Lp | BV | BVp | ND | NDp | AE | AEp |
|---|---|---|---|---|---|---|---|---|
| $z \to +\infty$ | $E \to 0$ | $E \to \infty$ | $E \to \infty$ | $E \to 0$ | $E \to 0$, $k_x < 1/2H$ $E \to \infty$, $k_x > 1/2H$ | $E \to 0$, $k_x > 1/2H$ $E \to \infty$, $k_x < 1/2H$ | $E \to 0$, $k_x > 1/2H$ $E \to \infty$, $k_x < 1/2H$ | $E \to 0$, $k_x < 1/2H$ $E \to \infty$, $k_x > 1/2H$ |
| $z \to -\infty$ | $E \to \infty$ | $E \to 0$ | $E \to 0$ | $E \to \infty$ | $E \to \infty$, $k_x < 1/2H$ $E \to 0$, $k_x > 1/2H$ | $E \to \infty$, $k_x > 1/2H$ $E \to 0$, $k_x < 1/2H$ | $E \to \infty$, $k_x > 1/2H$ $E \to 0$, $k_x < 1/2H$ | $E \to \infty$, $k_x < 1/2H$ $E \to 0$, $k_x > 1/2H$ |





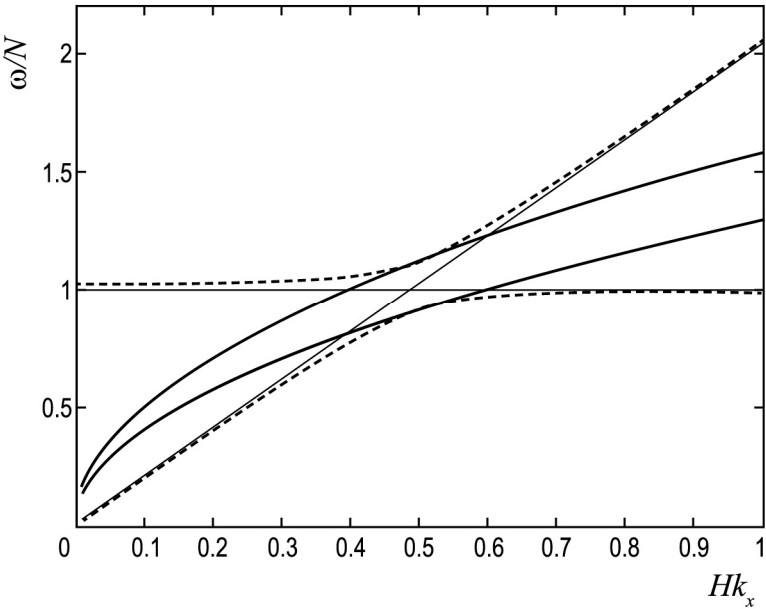

**Figure 1: Dispersion dependencies** $\omega(k_x)$ **: 1) boundaries of the free propagation of AGW: at the top - the acoustic area, at the bottom - the gravity area (dashed lines); 2) evanescent mode:** $\omega = \sqrt{k_x g}$ **(upper solid curve) and** $\omega = \sqrt{k_x g(\gamma - 1)}$ **(lower solid curve),** $\omega = N$ **(thin horizontal line),** $\omega = k_x c$ **(thin sloping straight line).**



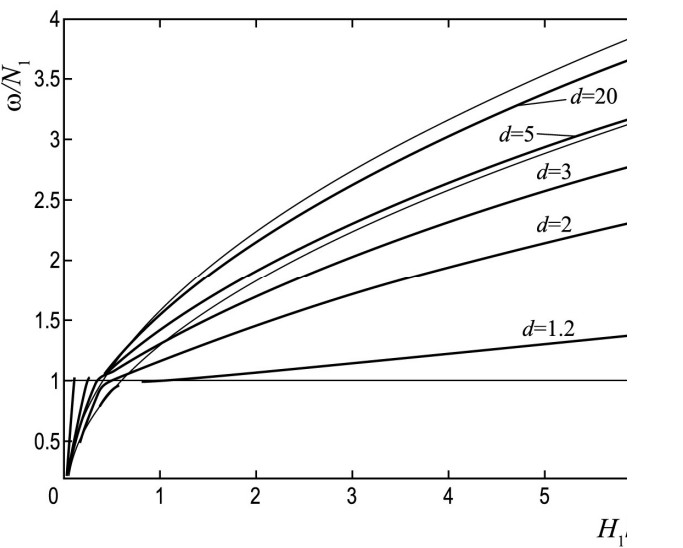
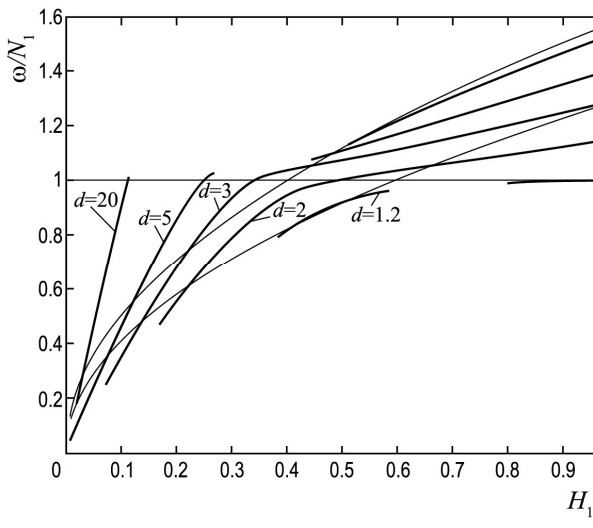

**Figure 2: Dispersion dependencies** $\omega = f(k_x)$ **at the boundary of the discontinuity for different values of the parameter** $d$**. General dependence (a), long-wave part in more detail (b). Thin curves denote** $\omega = \sqrt{k_x g}$ **(upper curve) and** $\omega = \sqrt{k_x g (\gamma - 1)}$
5    **(lower curve),** $\omega = N$ **(horizontal line).**





**Figure 3: Dispersion dependencies of the** $\omega = f(k_x)$ **type at the temperature discontinuity boundary for** $d = 2$ **(a),** $d = 3$ **(b),** $d = 5$ **(c),** $d = 20$ **(d). The dashed curves represent the boundaries of the areas with free propagation of AGW in the upper and** 5    **lower half-space. Thin solid curves represent dispersions** $\omega = \sqrt{k_x g}$ **(upper curve) and** $\omega = \sqrt{k_x g(\gamma - 1)}$ **(lower curve).**





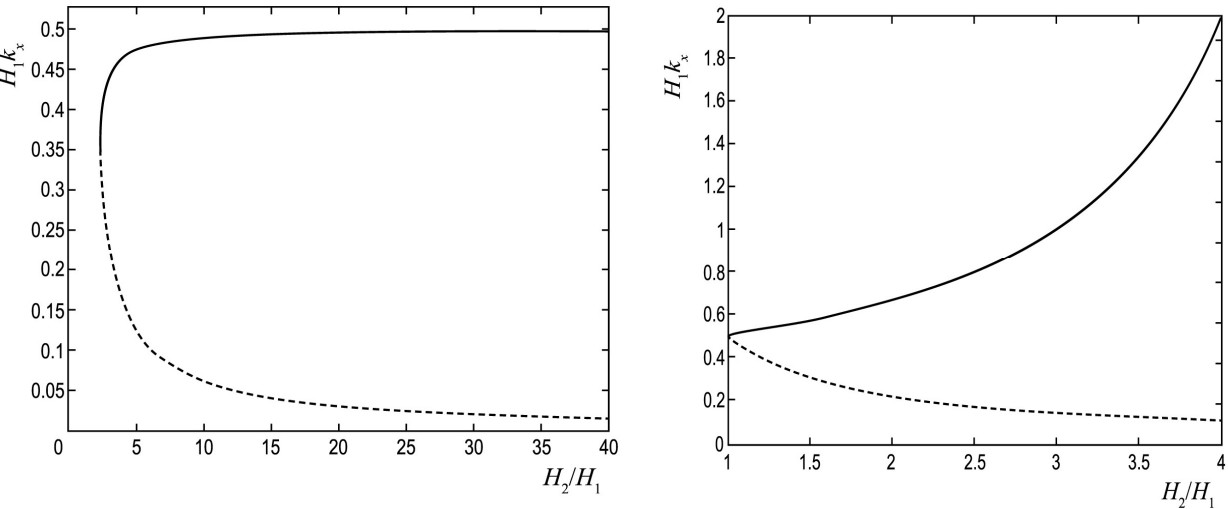

**Figure 4: Horizontal scales** $k_x H_1$, **on which the modes with the dispersion** $\omega^2 = k_x g$ **(a) and** $\omega^2 = k_x g (\gamma - 1)$ **(b) are realized, depending on** $d = H_2 / H_1$.

