# Peer review of "EVANESCENT ACOUSTIC-GRAVITY MODES IN THE ISOTHERMAL ATMOSPHERE: SYSTEMATIZATION, APPLICATIONS TO THE EARTH'S AND SOLAR ATMOSPHERES"

_Annales Geophysicae, 2019_

## Referee Comment (RC1) · Tamaz Kaladze (Referee) · 26 Feb 2019

In the present manuscript, the properties of so-called evanescent acoustic-gravity waves, which propagate only horizontally are investigated. The atmosphere is assumed to be isothermal and stratified by gravity. The possibility of existence of new types of acoustic-gravitational wave modes, not previously studied, is shown. Their pseudo-modes, which satisfy the same dispersion equation, but differ in polarization and height dependence of disturbance amplitudes, were also investigated. The results obtained may be useful for understanding the physics of wave processes in the atmo-

spheres of the planets. They can be used to explain wave observations in the Earth's atmosphere and on the Sun.

At the same time, there are a number of questions to the manuscript that need clarification:

1. The correctness of the transition from equations (1), (2) to equations (5), (6) when considering divergence-free waves (divV = 0) is in doubt. Since in an incompressible medium the speed of sound tends to infinity, the product "infinity to zero" in square brackets of equations (1), (2) becomes uncertain.

2. The realization of the obtained modes was considered in the framework of a simplified model of an infinitely thin discontinuity in the altitude profile of temperature. In the atmospheres of the planets, the situation is most likely realized when the change of the parameters along the vertical occurs on scales of tens of kilometers, or even hundreds, for the Sun. Can the acceptance of the finite thickness of the transition layer significantly affect the conditions of realization of the evanescent wave modes considered in the work?

3. In the atmosphere, different types of gravity disturbances may occur:

(1) freely propagating waves, having a real and non-zero vertical component of the wave vector; (2) evanescent wave modes, propagating only horizontally.

Does it mean that sources of evanescent modes and freely propagating waves are fundamentally different?

4. It is not clear from the manuscript how complete is the list of possible evanescent modes. Are there additional requirements for disturbances that will lead to new solutions?

The following inaccuracies or typographical errors are seen in the manuscript.

1) Based on the content of the manuscript, in the Abstract (lines 10-11 of page 1), the

phrase "the properties of the medium" should be replaced by "the properties of the disturbances".

2) The bottom line of Table 2 (line 2, p. 20) uses the abbreviations AMp, AM, which, apparently, should be replaced by AE and AEp.

3) Mistakes are made in the names of subsections 2 and 2.1 (line 1, page 3 and line 19, page 3). Probably should be

2 Evanescent modes in the isothermal atmosphere

2.1 Non-divergent and pseudo-non-divergent modes

As the conclusion I would like to emphasize that conducted in the manuscript research is very important for the relevant science and it may be published in Annales Geophysicae after necessary revision taking into account my comments.

---

## Author Comment (AC1) · 1 Mar 2019

Reply to the report by Reviewer_1

We would like to thank the Reviewer_1 for the questions and recommendations. Below we provide the answers to the proposed questions.

Reviewer 1:

1 The correctness of the transition from equations (1), (2) to equations (5), (6) when

considering divergence-free waves (divV = 0) is in doubt. Since in an incompressible medium the speed of sound tends to infinity, the product "infinity to zero" in square brackets of equations (1), (2) becomes uncertain.

We consider a compressible atmosphere stratified in a field of gravity. In equations (1), (2), the speed of sound refers to a compressible medium and is the final value, whereas the condition divV = 0 determines the properties of the perturbations only.

2. The realization of the obtained modes was considered in the framework of a simplified model of an infinitely thin discontinuity in the altitude profile of temperature. In the atmospheres of the planets, the situation is most likely realized when the change of the parameters along the vertical occurs on scales of tens of kilometers, or even hundreds, for the Sun. Can the acceptance of the finite thickness of the transition layer significantly affect the conditions of realization of the evanescent wave modes considered in the work?

The simplest model of a thin temperature gap is considered as an example in order to show the fundamental possibility of implementing the new types of wave modes obtained in the work. To understand how the thickness of the transition layer affects the properties of the modes considered, a separate study is needed. This effect seems to be significant. Especially when the magnitude of the transition layer is commensurate with the atmosphere scale height for the upper or lower isothermal half-spaces. In our opinion, it is more expedient to investigate the implementation of the received modes within the framework of an atmospheric model with a continuous non-isothermal altitude profile of temperature.

3. In the atmosphere, different types of gravity disturbances may occur: (1) freely propagating waves, having a real and non-zero vertical component of the wave vector; (2) evanescent wave modes, propagating only horizontally. Does it mean that sources of evanescent modes and freely propagating waves are fundamentally different?

The problem of sources was not analyzed in the work, we considered free waves (on

the right side of equations (1), (2) there are zeros). It is unlikely that the sources of atmospheric acoustic-gravity waves of different types must necessarily have a different nature. In our opinion, if the sources are localized in the isothermal interval of the heights of the atmosphere, they generate a freely propagating AGW more effectively. At heights of sharp temperature gradients, evanescent wave modes are preferred, since "surfaces" arise that support the propagation of such waves. This question requires separate study.

4. It is not clear from the manuscript how complete is the list of possible evanescent modes. Are there additional requirements for disturbances that will lead to new solutions?

It is likely that equations (1), (2) admit the possibility of the existence of other types of evanescent wave modes. When imposing other additional conditions on the properties of disturbances, besides those considered in the article, other types of evanescent modes can be obtained and Table 1 can be supplemented.

The following inaccuracies:

1) The phrase "the properties of the medium" should be replaced by "the properties of the disturbances".

2) The bottom line of Table 2 (line 2, p. 20) uses the abbreviations AMp, AM, which, apparently, should be replaced by AE and AEp.

3) Mistakes are made in the names of subsections 2 and 2.1 (line 1, page 3 and line 19, page 3). Probably should be 2 Evanescent modes in the isothermal atmosphere 2.1 Non-divergent and pseudo-non-divergent modes

We have checked and corrected these inaccuracies in the text.

Sincerely, Authors
* * *

---

## Referee Comment (RC2) · Anonymous Referee #2 · 10 Mar 2019

The paper presents a comprehensive study of acoustic-gravity waves in a stratified atmosphere. This research topic has important applications in astrophysics and geophysics. In general, the paper is well-written, but I believe it would benefit from resolving the issues summarised below:

The title: Perhaps, the authors should modify the title of the paper, as it addresses also the case of a vertically non-isothermal atmosphere.

p. 1, l. 23: ".. consisting of acoustic and gravity regions" - are those regions on the

dispersion plane or in different parts of the atmosphere?

p. 2, l. 30: the authors claim that the possibility of the existence of a new type of evanescent acoustic-gravity modes is proved in the paper. Could the authors explain why this mode has been missing from the vast amount of previous studies of this problem? In other words, which novel element (assumption or method) allowed the authors to identify this previously unknown mode.

p. 3, l. 8: Please mention that the sound speed is determined by the temperature.

Eq. (11): the RHS of the equation may be confusing: it is not clear that it actually consists of two different lines corresponding to different signs on the LHS. Please modify the equation, by, e.g., adding a comma after $k\_x$ in the top raw, and a full stop after $k\_x$ in the bottom raw.

It would be instructive to link the term "anelastic" with the terms "compressive" or "incompressive", which are commonly used in the solar atmospheric research.

The term "an unlimited atmosphere" would perhaps sound better as "an unbounded atmosphere".

p. 10, l. 8: Please give the physical meaning of this boundary condition. In other words, the continuity of which physical quantity or quantities should be kept across the interface?

Throughout the paper: please use "equation" instead of "equality".

p. 11, l. 1: It is not clear how the 8th order polynomial in Eq. (31) is obtained from Eq. (30) which has a 4th order polynomial in the numerator.

p. 15, l. 4: "the f-mode observed on the Sun should not be compared with the non-divergent ND mode, but with non-divergent pseudo–mode NDp." First of all, I think that the word "associated" would be better than "compared" in this context. Anyway, please explain the physical implications of this association (or comparison).

Table 2 and 3: Please remind the abbreviations used in the tables (i.e., "L", "Lp", "BV", "BVp", etc.) in the captions. It would allow using those tables in review papers and presentations.
* * *

---

## Author Comment (AC2) · 18 Mar 2019

**Reply to Referee_2**

We would like to thank the Referee_2 for the recommendations that have helped us to improve our manuscript. Below we provide answers for your questions.

*Referee 2:*

*The title: Perhaps, the authors should modify the title of the paper, as it addresses also the case of a vertically non-isothermal atmosphere.*

In the first part of the work, we considered.separate types of modes in an unlimited isothermal atmosphere. In the second part, we studied the possibility of realizing the modes at a temperature discontinuity, but for each half-space within the isothermal model. Therefore, we are of the opinion that the term "isotermic atmosphere" in the title is appropriate.

*p. 1, l. 23 "consisting of acoustic and gravity regions" - are those regions on the dispersion plane or in different parts of the atmosphere?*

We changed the sentence to «consisting of acoustic and gravity regions on the dispersion plane».

*p. 2, l. 30: the authors claim that the possibility of the existence of a new type of evanescent acoustic-gravity modes is proved in the paper. Could the authors explain why this mode has been missing from the vast amount of previous studies of this problem? In other words, which novel element (assumption or method) allowed the authors to identify this previously unknown mode.*

We assumed that wave disturbances may exist in a stratified compressible atmosphere that satisfy the new additional conditions. Under the assumption of perturbation incompressibility ($divV = 0$), the known ND mode was obtained, and under the assumption of perturbation inelasticity ($div(\rho_0 V) = 0$) a new AE mode was obtained. In the text of the article indicated, under which assumption each of these modes is obtained.

*p. 3, l. 8: Please mention that the sound speed is determined by the temperature.*

We added in the text the definition of the atmospheric scale height in the form $H = kT / mg$, clearly indicating the dependence on temperature.

*Eq. (11): the RHS of the equation may be confusing: it is not clear that it actually consists of two different lines corresponding to different signs on the LHS. Please modify the equation, by, e.g., adding a comma after k_x in the top raw, and a full stop after k_x in the bottom raw.*

Equation (11) was written in one line.

*It would be instructive to link the term "anelastic" with the terms "compressive" or "incompressive", which are commonly used in the solar atmospheric research.*

According to the physical meaning, ND mode is "incompressible" ($divV = 0$), the other considered modes, that is, NDp, AE, AEp modes, are "compressible" ($divV \neq 0$). We use the term "anelastic" for disturbances with $div(\rho_0 V) = 0$.

*The term "an unlimited atmosphere" would perhaps sound better as "an unbounded atmosphere".*

Replaced.

*p. 10, l. 8: Please give the physical meaning of this boundary condition. In other words, the continuity of which physical quantity or quantities should be kept across the interface?*

Obviously, if the atmosphere is barometric, then the equilibrium pressure $p_0$ should be continuous across the interface, and hence the value of $\rho c^2$ also. In addition, for the perturbed values, we require continuity of the vertical velocity component $V_z$ (kinematic condition) and perturbed pressure (dynamic condition). Under these assumptions, we obtained equations (29), (30). For more details see, for example, Tolstoy (1963), Rosental and Gough (1994), Cheremnykh et al. (2018a).

*Throughout the paper: please use "equation" instead of "equality".*

Corrected.

*p. 11, l. 1: It is not clear how the 8th order polynomial in Eq. (31) is obtained from Eq.(30) which has a 4th order polynomial in the numerator.*

To get rid of the radicals in expressions (28), (29), which determine the values of a1 and a2, these expressions were squared several times when a polynomial was obtained. The procedure for obtaining a polynomial from boundary conditions (30) is described in Miles and Roberts (1992). Note that expression (31) is only part of the full polynomial expression obtained in Miles and Roberts (1992). Moreover, in (31) two non-physical roots are omitted.

*p. 15, l. 4: "the f-mode observed on the Sun should not be compared with the nondivergent ND mode, but with non-divergent pseudo–mode NDp." First of all, I think that the word "associated" would be better than "compared" in this context. Anyway, please explain the physical implications of this association (or comparison).*

We agree that "associated" better reflects the meaning of the statement. ND mode and pseudo – mode NDp have the same variance. Therefore, when observing $\omega$ (kx), these modes are indistinguishable. The modes differ in the sign of polarization (in one mode, the Vx oscillations are ahead of Vz by 90 °, and in the other mode they are 90 ° behind) and the pattern of amplitude variation with height. Physically, ND mode is "incompressible" (divV = 0), and NDp mode is "compressive" (divV $\neq$ 0). In the framework of the considered model, only the NDp mode can satisfy the condition of energy reduction in both sides of the interface if the temperature in the upper half-space is higher than in the lower half-space.

*Table 2 and 3: Please remind the abbreviations used in the tables (i.e., "L", "Lp", "BV", "BVp", etc.) in the captions. It would allow using those tables in review papers and presentations.*

We gave the full names of the modes in the headings of Tables 2 and 3.

Sincerely,
On behalf of the authors of the manuscript,

Yuriy Selivanov, PhD

---

## Referee Comment (RC3) · Anonymous Referee #2 · 20 Mar 2019

The authors managed to take satisfactorily into account all my comments. I am happy to recommend the acceptance of the manuscript.

---

## Referee Comment (RC4) · Tamaz Kaladze (Referee) · 29 Mar 2019

Dear Editor, I am pleased to answer you that the authors addressed my comments in very nice fashion and I agree that the manuscript should be published. Thanks

---

## Editor Comment (EC1) · Sergiy Shelyag (Editor) · 29 Mar 2019

Dear Tamaz,

could you please advise the authors on whether you would like to continue the discussion or accept the answers?

Thanks Best regards
* * *

---

## Editor Comment (EC2) · Sergiy Shelyag (Editor) · 29 Mar 2019

Dear Tamaz,

thanks for that and your very quick reply.

Best regards

---

## Author Comment (AC3) · 2 May 2019

The authors thank the Reviewer for his valuable comments, which led to the improvement of the discussed manuscript.
* * *

---

## Author Comment (AC4) · 2 May 2019

The authors thank the Reviewer for his valuable comments, which led to the improvement of the discussed manuscript.
* * *